# Suitability of Nanoparticles to Face Benzo(a)pyrene-Induced Genetic and Chromosomal Damage in *M. galloprovincialis*. An In Vitro Approach

**DOI:** 10.3390/nano11051309

**Published:** 2021-05-15

**Authors:** Margherita Bernardeschi, Patrizia Guidi, Mara Palumbo, Massimo Genovese, Michela Alfè, Valentina Gargiulo, Paolo Lucchesi, Vittoria Scarcelli, Alessandra Falleni, Elisa Bergami, Francesca S. Freyria, Barbara Bonelli, Ilaria Corsi, Giada Frenzilli

**Affiliations:** 1Section of Applied Biology and Genetics and INSTM Local Unit, Department of Clinical and Experimental Medicine, University of Pisa, 56126 Pisa, Italy; margherita.bernardeschi@for.unipi.it (M.B.); patrizia.guidi@unipi.it (P.G.); m.palumbo@studenti.unipi.it (M.P.); paolo.lucchesi@unipi.it (P.L.); vittoria.scarcelli@unipi.it (V.S.); alessandra.falleni@unipi.it (A.F.); 2Department of Experimental and Clinical Biomedical Sciences “Mario Serio”, University of Florence, 50121 Florence, Italy; massimo.genovese@student.unisi.it; 3Institute of Science and Technology for Sustainable Energy and Mobility STEMS-CNR, 80126 Naples, Italy; michela.alfe@stems.cnr.it (M.A.); valentina.gargiulo@stems.cnr.it (V.G.); 4Earth and Environmental Sciences and INSTM Local Unit, Department of Physical, University of Siena, 53100 Siena, Italy; bergami@student.unisi.it (E.B.); ilaria.corsi@unisi.it (I.C.); 5INSTM Unit of Torino-Politecnico, Department of Applied Science and Technology, 10129 Politecnico di Torino, Italy; francesca.freyria@polito.it (F.S.F.); barbara.bonelli@polito.it (B.B.)

**Keywords:** DNA damage, chromosomal damage, cytome assay, titanium dioxide nanoparticles, carbon black derived nanoparticles, genotoxicity, cellular uptake, mussel gill biopsy, nanoremediation, benzo(a)pyrene

## Abstract

Benzo(a)pyrene (B(a)P) is a well-known genotoxic agent, the removal of which from environmental matrices is mandatory, necessitating the application of cleaning strategies that are harmless to human and environmental health. The potential application of nanoparticles (NPs) in the remediation of polluted environments is of increasing interest. Here, specifically designed NPs were selected as being non-genotoxic and able to interact with B(a)P, in order to address the genetic and chromosomal damage it produces. A newly formulated pure anatase nano-titanium (nano-TiO_2_), a commercial mixture of rutile and anatase, and carbon black-derived hydrophilic NPs (HNP) were applied. Once it had been ascertained that the NPs selected for the work did not induce genotoxicity, marine mussel gill biopsies were exposed in vitro to B(a)P (2 μg/mL), alone and in combination with the selected NPs (50 µg/mL nano-TiO_2_, 10 µg/mL HNP). DNA primary reversible damage was evaluated by means of the Comet assay. Chromosomal persistent damage was assessed on the basis of micronuclei frequency and nuclear abnormalities by means of the Micronucleus-Cytome assay. Transmission Electron Microscopy (TEM) was performed to investigate the mechanism of action exerted by NPs. Pure Anatase n-TiO_2_ was found to be the most suitable for our purpose, as it is cyto- and genotoxicity free and able to reduce the genetic and chromosomal damage associated with exposure to B(a)P.

## 1. Introduction

Polycyclic aromatic hydrocarbons (PAHs) represent a large class of organic pollutants that are present in all environmental matrices, deriving mainly from the wide use of fossil fuel combustion and industrial processes [1]. Due to their persistence, semivolatility, easy accumulation and potential biomagnification in organisms, pollution from PAHs is a cause of great concern [2,3]. Among them, benzo(a)pyrene, (B(a)P) a highly lipophilic compound, is considered to be representative of the class of PAHs in organic pollutants [4] such as petroleum and crude oils. B(a)P is commonly detected in the marine environment where it comes in contact with biota, because of either acute or chronic exposure, exerting toxicity at different biological levels, resulting in consequences such as reproductive dysfunction, immune system destruction, neuro and endocrine toxicity, mutagenic response, tumor induction [3,5,6]. Within the specific heritable genetic alterations induced by B(a)P cell mutations, mutagenic-mediated mortalities and/or fertility reduction have been reported [7,8]. Moreover, dietary exposure to B(a)P has been shown to contribute to the development of human cancer [9], with B(a)P being classified by the IARC as a human carcinogen (Group 1). For all these reasons, B(a)P environmental concentrations are closely monitored [10], and B(a)P removal is essential for both the recovery of different environmental matrices and for wildlife and human health preservation. Thus, remediation of PAH-contaminated sites has been considered a great challenge during the last 20 years, although data from the literature are still scarce on aquatic environments, mainly with respect to soil remediation [11]. During the last decade, innovative remediation approaches have promoted the use of nanoparticles (NPs) and nanomaterials (NMs) to clean contaminated environmental matrices [12,13,14], and nanoremediation has been considered for the removal of organic contaminants due to successful results in remediating several pollutants [11]. Thanks to their unique physical-chemical properties, particularly their small size and their large surface area, NPs are employed in many different fields [15]. In fact, being more widely distributed in comparison to larger-sized particles, NMs can be considered more suitable for in situ remediation processes [16].

For this reason, a focus of future research could be the development of rapid, reliable, low cost and low risk-based PAHs cleanup strategies through the use of NPs. In this context, laboratory studies aiming to synthetize harmlessness NPs with respect to the biota, to be employed in in vivo and in vitro models, are necessary before any application in polluted aquatic environments. As TiO_2_ NPs were recently shown to adsorb several PAHs from soil and water [17], the potential of TiO_2_ NPs to protect against B(a)P-induced toxicity in biological systems appears to be of extreme interest, and that’s where the originality of this work lies. In the present study, TiO_2_ NPs both of commercial origin and specifically designed to interact with PAHs were selected. In addition, carbon black (CB)-derived hydrophilic NPs (HNP) were designed and specifically synthetized for the same purpose. These nanoparticles were selected because of their different elemental compositions (inorganic and organic NPs) and for their high dispersibility in the exposure media. Taking into consideration B(a)P genotoxicity, it is well recognized that DNA integrity is a fundamental requirement for preventing mutagenic and carcinogenic events, which can occur if DNA damage is not properly repaired.For this reason, cellular and molecular genotoxicity biomarkers have been widely used as early warning signals in environmental mutagenesis [18,19]. In the present study, DNA primary damage, a biomarker of exposure, and chromosomal damage, a biomarker of effect, were used. In particular, DNA primary damage was assessed by Comet assay, while chromosomal damage including the presence of micronucleated cells, nuclear morphology alterations and apoptotic cells was evaluated by Cytome assay. The Comet assay is used to microscopically detect primary repairable, and therefore reversible, DNA lesions (DNA single strand breaks, double strand breaks, alkali labile sites) at the level of a single cell, providing an instantaneous picture of DNA primary damage. On the other hand, the Cytome assay gives an efficient measure of chromosomal damage detecting persistent DNA lesions or aneugenic effects, as a result of either chromosome breakage or chromosome mis-segregation during mitosis, that cannot be repaired [20]. Combining these two assays, which have such differences in terms of sensitivity, the endpoints measured, and the type of data, the evaluation of genotoxicity is significantly improved [21]. Looking for non-genotoxic NPs to be used for co-exposure experiments, a deeper investigation into the mechanisms underlying the potential genotoxicity of NPs and their interaction with cells, cellular uptake and particle behavior in exposure media was also carried out.

As the experimental model to be used, marine bivalves such as the *Mytilus* species have a long history of being used in biomonitoring and field studies due their widespread distribution and abundance, high filtration rate, and ability to bio-concentrate contaminants [22,23,24]; they are suspension feeders, and possess highly developed endocytic and phagocytic mechanisms, which can lead to the uptake of particles [23,25,26].

Many studies on the potential (geno)toxicity on fresh and marine water bivalves have been performed in vitro on hemocytes [27,28], even though it has been reported that in these mollusks, B(a)P genotoxicity was expressed earlier in gill cells [29], which were found to be more responsive to contaminants than haemolymph also in the case of monitoring studies (in situ exposure) and in vivo laboratory exposure [29,30]. For mollusk bivalves, gills represent a fundamental barrier against contaminants [31,32,33], being the first site of contact between the animal and the aquatic media, through which mutagens and carcinogens can be internalized and metabolized into reactive products [31,34,35].

The in vitro methodology presented in this study, i.e., the exposure of portions of the marine mussel gill tissue [36,37], makes it possible to overcome difficulties in marine mussel primary cell culture growing related to the lack of knowledge regarding specific growth factors [38,39,40] and the maintenance of sterile cell culture associated with the use of antibiotics [41]. The choice of the present experimental model was also due to the fact that direct exposure of gill biopsies freshly isolated from the animal makes it possible at the same time to (1) avoid long-term maintenance of the aquaria needed to host the specimens, (2) mimic the response to genetic insult after shorter times of exposure while still obtaining reliable data, and (3) reduce the chemical volumes needed to perform the experiments [42,43]. A link between exposure and associated molecular alterations is well known for in vitro studies, as well as the great potential in predicting in vivo chemical carcinogenicity [44,45]. For this reason, in the present study, portions of the marine mussel *Mytilus galloprovincialis* gill tissue were exposed for the first time to B(a)P in the presence of the selected NPs to simulate the interactions between cells and xenobiotics. Once it had been ascertained that the NPs selected for the work did not induce genotoxicity, their capacity to recover B(a)P-induced genotoxic effects was investigated.

The double aim of our work was to investigate the harmlessness of the selected NPs (nano-TiO_2_, in two crystalline forms, and HNP), as well as their capability to recover B(a)P-induced genotoxic effects. The novelty of this study lies in the application of specifically designed nanoparticles to remediate the genetic and chromosomal damage induced by B(a)P in *Mytilus galloprovincialis* gill biopsies. The chosen biological target is also peculiar, since it mimics the route of exposure of the whole animal, being a sort of bridge between the cellular and the organism level.

In the present study, the good performance exhibited by one of the two formulations of nano-TiO_2_ in lowering the genotoxic insult exerted by B(a)P, and without inducing any additional damage, was highlighted. 

## 2. Materials and Methods 

### 2.1. Chemicals and Devices

Low Melting Agarose (LMA), Normal Melting Agarose (NMA), Dispase, Ethidium Bromide (EtBr), Triton X−100, Ethylenediaminetetra acetic acid (EDTA), Trizma Base, Dimethyl Sulfoxide (DMSO), NaOH, Tris-HCl, Trypan blue, Acetic acid, Epon Araldite components, and sodium cacodylate were purchased from Sigma-Aldrich (Steinheim, Germany). HBSS and lead citrate constituents were purchased from Carlo Erba Reagenti (Milan, Italy). Ethanol and HCl were purchased from PanReac AppliChem (Barcelona, Spain). Giemsa was purchased from Titolchimica spa (Rovigo, Italy). Artificial seawater (ASW) constituents were purchased from Sigma-Aldrich (Milan, Italy). Karnovsky fixative components, osmium tetroxide, uranyl acetate, and propylene oxide were purchased from Fisher Scientific Italia (Milan, Italy). Mesoporous n-TiO_2_ (25 nm), namely MT, was synthetized at Politecnico di Torino (Turin, Italy), by following a published procedure [46] and by using ACS (American Chemical Society)-grade chemicals from Sigma-Aldrich (Milan, Italy). n-TiO_2_ (25 nm), namely Aeroxide^®^ TiO_2_ P25 (P25) was kindly supplied by Eigenmann & Veronelli, Milan (Italy). To produce HNPs, carbon black (furnace type, N110 according to the ASTM classification) was obtained from Sid Richardson Carbon Co. HNP were produced at CNR-STEMS, Napoli (Italy) following a published procedure [47,48] and using ACS-grade chemicals from Sigma-Aldrich (Milan, Italy). An ultrasonic homogenizer HD 2070 was purchased from Bandelin Electronic (Berlin, Germany). Reichert-Jung Ultracut E ultramicrotome was purchased from Reichert Optiche Werke AG (Vienna, Austria). Trasmission Electron Microscopes: Tecnai G2 Spirit, (Hillsboro, OR, USA); JEOL 100 SX, JEOL ltd 3−1-2 Musashino (Tokyo, Japan). An AMT xr80b ccd camera was purchased from Woburn (MA, USA), and was used to perform particle characterization and analyze cellular uptake. A Dynamic Light Scattering Zetasizer Nano-ZS was purchased from Malvern Instruments (Worcestershire, UK). A centrifuge was purchased from Hettich (Tuttlingen, Germany). An X’Pert Philips PW3040 diffractometer was purchased from PANalytical (Almelo, The Netherlands). A Quantachrome Autosorb 1C was purchased from Boyton Beach (FL, USA). A pH meter was purchased from XS Instruments, (Modena, Italy). Fluorescent and Optical microscopes were purchased from Zeiss (Milan, Italy).

### 2.2. Particle Production and Characterization 

HNPs were produced in accordance with the synthetic approach recently described [48], and briefly summarized here. An amount of 500 mg of CB was treated with 10 mL of concentrated nitric acid (67 wt.%) at 100 °C under stirring for 24 h. The black suspension was cooled and centrifuged, and the recovered solid was washed with distilled water until complete removal of any acid traces. After that, the solid was dried at 100 °C at atmospheric pressure, and its main characteristics were checked to ensure the reproducibility of its typical chemico-physical characteristics. HNP presents an H/C ratio of 0.15 and an oxygen content of around 30 wt.%, in the form of carboxylic acid, lactone, lactol, and phenol groups [48]. MT (mesoporous titania) was synthetized by following a published procedure [46]. MT was characterized as previously described [49,50], using XRD (X-ray diffraction) powder diffraction at 40 kV and 40 mA equipped with Cu Kα radiation (step scan = 0.02 2θ, time per step = 2 s) and N_2_ adsorption/desorption isotherms at −196 °C. Before N_2_ adsorption, the powder was outgassed at 150 °C for 4 h to remove water and other atmospheric contaminants). Sample specific surface area (SSA) was calculated according to the Brunauer–Emmett–Teller (BET) method; pore total volume was measured at p/p^0^ = 0.99. 

The resulting MT powder was characterized in the media used for in vitro study in filtered (0.22 µm) ASW (10 mg mL^−1^) prepared according to the protocol ASTM D1141–98 [46,51] (32‰ salinity, pH 8 ± 0.1). MT stock suspension in ASW (10 mg mL^−1^) was probe-sonicated for 45 min (100 W, 50% on/off cycle), diluted to working suspension in ASW, and again sonicated for 15 min prior to use. 

The P25 (declared purity of 99.9%) characterization protocol has been reported previously [50]. Nano-TiO_2_ stock suspension was prepared in filtered (0.22 µm) ASW (10 mg mL^−1^) (32‰ salinity, pH 8 ± 0.1), probe-sonicated for 45 min (100 W, 50% on/off cycle), and immediately diluted to a working suspension, always in ASW, which was again sonicated for 15 min prior to use in accordance with the previously reported protocols [37]. 

### 2.3. Particle Behavior

Particle behavior in exposure media was fully characterized by dynamic light scattering (DLS). DLS measurements of MT and P25 at 10 mg L^−1^ were performed in ASW (0.22 µm) by means of hydrodynamic size (Z-average, nm) and Polydispersity Index (PDI) using a Zetasizer Nano Series software, version 7.02 (Particular Sciences, UK). Measurements were performed in triplicate, at 23 ± 13 °C, each consisting of 11 runs of 10 s for NP size-related parameters. In the case of HNP, the zeta potential measurements were performed on HNP aqueous suspension (double-distilled water) at a concentration of 0.05 mg mL^−1^ as a function of pH (each point of pH was reached by adding 100 µL of 0.1 M NaOH to HNP aqueous suspension and then by subsequent additions of increasing amounts of 0.1 M HCl). 

MT and P25 primary size and morphology were addressed by means of Transmission electron microscopy (TEM) through a Tecnai G2 Spirit operating at 100 kV. Both MT and P25 were dispersed in Milli-Q, and upon 24 h incubation in ASW at a concentration of 10 mg L^−1^, a 10 µL-drop of each suspension was placed on a formvar/carbon-coated copper grid and dried before imaging. MT and P25 suspended in ASW were extensively washed before deposition and drying, while Milli-Q dispersions were straightforwardly imaged. Given the chemical composition of ASW, a washing procedure was required to reduce the overall content of salts, which otherwise would have been crystallized during sample drying yielding images of poor quality. Briefly, after incubation in ASW, MT and P25 were pelleted at 18 rcf at 4 °C for 20 min and resuspended in Milli-Q. The process was repeated three times. Finally, the newly achieved suspensions were dried and analyzed by TEM. 

### 2.4. In Vitro Exposure

Adult marine mussels of *M. galloprovincialis* (with a shell length range of 50–75 mm) were collected from a commercial mussel farm at Arborea (Orestano Gulf, Sardinia. UNI EN ISO 9001:2015. N°8143 Certiquality). After collection, mussels were transported to the laboratory and immediately sacrificed to collect gill biopsies. Three animals for each experimental point, and three replicates for each treatment (n = 9) were used, and at least five biopsies were obtained from each individual organism. Gills were placed into a 24-well plate, and for each animal, five different treatments were performed: control (C), positive control (C+, H_2_O_2_ 100 µM), exposure to NPs (50 µg/mL nano-TiO_2_ (MT or P25), 10 µg/mL HNP), exposure to B(a)P (2 μg/mL), and co-exposure. Stock suspensions of MT and P25 were freshly prepared and sonicated for 30 min at 35 KHz immediately before use. HNP suspensions were also freshly prepared immediately before use, but avoiding sonication due to the high solubility of the particles [48]. The stock suspensions were used undiluted for the highest exposure dose; all other doses were obtained by dilution. The experimental time was 1 h, at +4 °C and in the dark. Experiments were conducted in triplicate. 

### 2.5. DNA Primary Damage (Comet Assay) 

To investigate DNA primary damage, the Single Cell Gel Electrophoresis (SCGE or Comet assay) was used as described by Nigro and co-workers [52]. Among the various versions of the assay, the alkaline method (pH > 13) identifies the broadest spectrum of DNA damage, being able to detect double strand breaks (DSB), single strand breaks (SSB), alkali labile sites (ALS) that are expressed as single strand breaks, and single strand breaks arising as DNA repair intermediates. After dissection, gills were put in 5 mL HBSS 20‰ for 30 min, then transferred to a dispase/HBSS solution at 37 °C for 20 min. After digestion, the enzyme was inactivated using cold 20‰ HBSS solution. The resulting digestion product was filtered through a 100 μm mesh nylon filter. The cell suspension obtained was centrifuged at 125× *g* for 5 min and the pellet used for both the Comet and the Cytome assays. For the Comet assay, the experimental procedure was performed under yellow light, to avoid additional DNA damage. The pellet was re-suspended in HBSS, and 100 µL of the cell suspension was centrifuged at 125× *g* for 10 min. The resulting pellet was mixed with 75 µL of 0.5% LMA in calcium and magnesium free buffered saline (PBS), and spread on conventional slides, previously covered with a layer of 1% NMA. After the cell-containing agarose was polymerised (5 min on metal tray over ice), a final layer of 85 µL of LMA was added. Following agarose solidification, slides were lowered into freshly made lysing solution (2.5 M NaCl, 10 mM Tris, 0.1 M EDTA, 1% Triton X−100, and 10% DMSO, pH 10) for at least 1 h. Slides were placed in a horizontal gel electrophoresis chamber and covered for 10 min with fresh electrophoresis buffer (0.075 M NaOH, 1 mM EDTA, pH 13) to allow DNA unwinding. Electrophoresis was run at 25 V, 300 mA, for 5 min at pH 13. After the electrophoresis, slides were washed three times (for 5 min each) with Tris-HCl (pH 7.5) to neutralize the pH and allow DNA staining. Slides were stained with ethidium bromide and observed under a fluorescence microscope (400×). Damaged nuclei were comet shaped due to DNA migration towards the anode. The amount of DNA damage was evaluated as the percentage of DNA migrating out of the nucleus by an image analyzer (Komet 5.0 Software, Kinetic Imaging Ltd.), connected to the fluorescent microscope (Appendix A). Tail DNA (%) was chosen as a reliable Comet assay parameter. The fields on the microscope slide were visually randomly identified, two slides per mussel were set up, 50 random nuclei per slide were scored, and the mean was calculated. 

### 2.6. Chromosomal Damage (Cytome Assay) 

Micronucleated cells and morphological nuclear abnormalities frequencies were evaluated by Cytome assay. Aliquots of the mussel gill cell pellets were prefixed for 20 min in a solution containing 5% acetic acid, 3% ethanol, 92% HBSS 20‰, and centrifuged for 5 min at 475× *g*. The supernatant was removed, and 5 mL of fixative solution (from 5:1 to 3:1, depending on the humidity) was added to the suspended pellet; this process was repeated twice. After the last fixation and centrifugation, suspended cells were spread on slides (two slides per mussel), air dried, and stained with 5% Giemsa solution for 10 min. One thousand cells with preserved cytoplasm (Appendix A) per specimen were scored (500 per slide) to determine the frequency of micronuclei (MN), total nuclear abnormalities (NA) (which include nuclear blebs, nuclear buds, notched nucleus, nucleoplasmic bridges (NPB), circular nucleus, lobed nucleus) and apoptotic cells (APO), according to the criteria stated by Fenech [53].

### 2.7. Uptake of NPs 

Transmission Electron Microscopy (TEM) analyses were performed as described by Guidi and co-workers [54], with slight modifications. Briefly, tissues exposed for 1 h were washed to remove powders and treated with Karnovsky fixative for 5 h at room temperature, washed in 0.1 M cacodylate buffer overnight, postfixed in 1% aqueous osmium tetroxide for 2 h in the dark at room temperature, washed with distilled water, and dehydrated first in graded series of ethanol and then in pure propylene oxide. Samples were pre-embedded in Epon Araldite–propylene oxide 1:1 mixture overnight in slow rotation, followed by pure Epon Araldite for 6 h, and then embedded in new Epon Araldite at 60 °C for 48 h. Ultra-thin sections (70–90 nm) were cut using ultramicrotome Reichert-Jung Ultracut E, collected on a 200 mesh formvar carbon-coated copper grid and finally stained with 5% uranyl acetate and lead citrate. Samples were observed at 80 kV with a JEOL 100 SX transmission electron microscope equipped with an AMT XR80B ccd camera.

### 2.8. Statistical Analysis

Statistical analysis was performed using the software SGWIN (Windows 98). For genotoxicity data, Multifactor Analysis of Variance (MANOVA) was carried out by considering the dose, treatment, and experiment as independent variables. The Multiple Range Test (MRT) was applied in order to detect differences (*p* < 0.05) among different treatment groups. 

## 3. Results

### 3.1. Hydrophilic CB-Derived Nanoparticles (HNP), Aeroxide^®^ TiO_2_ P25 and Mesoporus Titania (MT) Characterization 

HNP, despite the strong oxidative conditions applied for their production, preserves the nanostructured features of the parent CB. TEM analysis (Figure 1A) confirmed that HNP consisted of aciniform agglomerates of almost spherical primary particles (average diameter of the primary particles: 15–20 nm). Z-potential values between −47 and −50 mV in the pH range 6–12 were indicative of a negative surface charge and of a good colloidal stability. Z-Average (nm) and the PDI in water are reported in Table 1. Thanks to their hydrophilic nature, the HNP do not undergo significant agglomeration in ASW (Table 1). 

A tendency for agglomeration of P25 in ASW was also observed by TEM (Figure 1B) and confirmed by DLS (Z-average of 900 nm). Overall, the DLS results highlighted how P25 NPs were prone to agglomerate in ASW (Table 1).

MT characterization performed by TEM (Figure 1C) showed elongated particles with rather uniform shape and dimension. The same sample was characterized by 12.4 ± 1.3 nm pure anatase crystallites (based on XRD technique, showing that the sample was 100% anatase) and very similar dimension nanoparticles with elongated shape and rather homogeneous size, forming agglomerates in the powder.MT sample mostly showed inter-particle mesopores, along with some (residual and) smaller intra-particle mesopores, finally leading to a SSA of 150 m^2^ g^−1^ and a pore volume of 0.39 cm^3^ g^−1^, as determined by N_2_ sorption isotherms at −196 °C. DLS analysis of MT suspensions at 10 mgL^−1^ in ASW showed the formation of large micron-scale agglomerates, with high Z-average values and very broad PDI compared to those in ultrapure water (MilliQ W) (Table 1). Sonication caused the breakage of agglomerates, but the particles quickly re-agglomerated, in particular in high ionic strength media such as ASW.

### 3.2. Cellular Uptake

The analysis through TEM highlighted how, in the biopsies treated with the three different NPs (Figure 2), it was possible to observe a consistent number of particles (indicated by arrows, Figure 2B–D) in comparison with the control (Figure 2A). Moreover, a general cytoplasmic rarefaction (lesser cytoplasmic density) was also observed in all the treated samples (Figure 2B–D) with respect to the control. In addition, *M. galloprovincialis* gill biopsy exposed to 10 μg/mL HNP showed large cytoplasmic vacuoles, and gill biopsy exposed to 50 μg/mL of MT exhibited altered mitochondria with matrix dilution and crystolisis. Occasionally, hypertrophic mitochondria were also visible in the control.

### 3.3. In Vitro Exposure 

#### 3.3.1. DNA Primary Damage 

Preliminary experiments were set up on gill biopsies to select different sub-genotoxic doses of NPs, displaying a cell viability higher than 90%, to be used for the co-exposure investigations with B(a)P (data not shown). In particular, the selected dose for HNP was 10 μg/mL, because the sub-genotoxic dose of 50 μg/mL was shown to be cytotoxic, and the highest tested dose (100 μg/mL) showed a genotoxic effect. For both of the nano-TiO_2_, even in the absence of genotoxic responses, doses higher than 50 μg/mL showed a certain degree of cell toxicity (Appendix A).

With respect to the co-exposure experiments, for the HNP treatments, after 1 h of exposure the NPs did not induce any statistically significant loss of DNA integrity with respect to the control. On the contrary, treatment with B(a)P provoked a statistically significant increase (*p* < 0.05) in DNA damage. With respect to co-exposure, even if a reduction of B(a)P-induced DNA damage was observed, it did not reach the control level (Figure 3A).

Treatment with P25 showed that NPs per se induced a loss (*p* < 0.05) of DNA integrity, even if it was lower than that induced by B(a)P, as revealed by the Multiple Range Test. The co-exposed cells showed levels of DNA primary damage comparable to the control (Figure 3B). 

The MT treatment did not exert any genotoxic effect. Moreover, in the co-exposed samples, the B(a)P-induced DNA damage was restored to the control level (Figure 3C), giving it the best performance. 

#### 3.3.2. Chromosomal Damage and Morphological Nuclear Abnormalities

Concerning chromosomal damage, a higher frequency of MN cells was observed after treatment with HNP with respect to the control (*p* <0.05) (Figure 4A), while no increases of NA were assessed. On the contrary, no statistically significant differences were observed between MN control levels and those of the cells treated with P25. MN frequencies were found to be higher when biopsies were co-exposed with B(a)P and n-TiO_2_ (Figure 4B). On the other hand, an increase (*p* < 0.05) of NA was induced by P25 (*p* < 0.05) (Figure 4E). MT induced comparable frequencies of MN and NA at all of the experimental points. Indeed, neither the MT NPs nor the co-exposure (MT + B(a)P) treatments showed any statistically significant difference with respect to the controls (Figure 4C,F). Moreover, MT and HNP did not seem to trigger any apoptotic process, while after P25 exposure, an increase of apoptotic cells was observed (data not shown).

## 4. Discussion

In the present study, selected NPs, both inorganic, in the form of two formulations of nano-scale titanium dioxide, and CB-based, in the form of HNP, were investigated to see if they exerted any genotoxic effects, since the first aim of the work was the identification of non-genotoxic NPs to be used in the second part of the research. Then, for the first time, their capability to reduce B(a)P-induced genotoxicity in *M. galloprovincialis* gill biopsies was assessed in vitro. 

The two n-TiO_2_-based powders (Aeroxide^®^ TiO_2_ P25, or P25, and Mesoporous TiO_2_, or MT) and HNP were specifically selected with the purpose of mitigating aromatic polycyclic hydrocarbon toxicity. The performances of P25 toward organic and inorganic classical pollutants have previously been evaluated on the basis of in vitro and in vivo studies [37,50,55] using the marine mussel as an experimental model, but they have never been assessed with respect to B(a)P. On the contrary, MT and HNP have never been investigated for this potential application. 

In vitro assay is a useful approach, as laboratory investigations allow more controlled conditions, so that the results are almost totally amenable to the effects of the tested chemical. Moreover, the suitability of in vitro testing strategies for predicting in vivo responses as well as potential exploration of adverse outcome pathways has been reported [41]. In the present work, the entire gill biopsy was exposed before the cells were dissociated; the originality of this approach lies in the exposure of a piece of metabolizing tissue that mimics the route of exposure of the whole animal, while still maintaining the characteristic controlled conditions of an in vitro study. Thus, this approach appears useful for better investigating the genotoxic potential of classical pollutants, as well as their interaction with nanoparticles, providing preliminary evidence regarding the possible scenario occurring in the environment. The combination of the alkaline version of the Comet assay with cyto-genetic tests, such as the Cytome assay, has been reported to be the most informative way to analyze the nano-genotoxic effects in bivalves. Indeed, the alkaline version of the Comet assay enables the identification of DNA single, double strand breaks and alkali labile sites, while the Cytome assay analyzes chromosomal damage induced by clastogenic (DNA breakage) or aneugenic (abnormal segregation) events in terms of micronuclei and nuclear abnormalities frequencies [56]. Moreover, in the present study TEM in cells was planned in order to check the actual internalization of NPs to verify that the potential genotoxic effects induced by NPs were paralleled by their cellular uptake. Generally, genotoxicity was always evaluated in the absence of cytotoxic effects with cell viability higher than 90%, as assessed by Trypan blue exclusion technique in order to avoid possible false positive results with the Comet assay. 

When analyzing the primary damage results, HNP did not itself exert a loss in DNA integrity. Concerning B(a)P co-exposure with HNP, a reduction of the genotoxicity caused by the contaminant was observed, but DNA integrity was not completely recovered to control levels. As shown in the co-exposure experiments, the results obtained with the Comet assay revealed quite high baseline levels of DNA primary damage ranging from 20% to 30% DNA migrated into the tail. Interestingly, such a low level of DNA integrity observed in the control specimens was also paralleled by ultrastructural alterations. As a recent work published by the same working group [14] actually showed a comparable DNA integrity value of the control mussels exposed in vivo, it can be speculated that the specimens used for the last two works, which came from the same farm in Sardinia, showed a certain degree of cellular damage at basal levels, even if this was still not high enough to guarantee the detection of statistically significant differences when compared with exposed animals. It is difficult to come to a definitive conclusion regarding the genotoxic potential of HNP through the Comet assay, since in absence of an agreed OECD guideline, different protocols have been applied to study the genotoxicity of carbon-based materials (carbon black (CB), CB-derived nanoparticles, CNT, and MWCNT, just to cite a few) in the pertinent literature (Appendix A). CBs are the carbon-based NPs closest to the HNP reported in this study, being their parent material. Anyway, it has to be underlined that the HNP chemical surface is quite different from that of the parent CB (parent CB is hydrophobic, while HNP is highly hydrophilic), despite having a quite similar morphology [48]. Taking into account the above-mentioned differences, and also keeping in mind that the term CB describes a class of differently nanostructured C-based materials with different surface characteristics (some CBs also bear on their surface adsorbedPAHs as production residuals), it is reasonable to use CB as a guideline to discuss HNP properties and behavior. Many studies have reported that CB also has a primary (direct or indirect) genotoxic effect on DNA [57], and few data are available regarding CB toxicity for bivalves, and these are limited to mussel hemocytes, where oxidative stress and ROS production occurs after in vitro and in vivo exposure [58]. On the other hand, it is also noteworthy that hydrophilic C-based NPs derived from a high CB destructuration can be regarded as toxicity safe in vivo, since they do not lead to any perturbations of different biological parameters on the vertebrate model *Danio rerio* [59]. 

Our data seem to confirm the low genotoxicity of HNP, which is likely related to indirect mechanisms such as HNP being arranged in large micrometric agglomerates. However, although HNP do not exert primary DNA damage by themselves at 10 µg/mL, they might not be suitable to reduce the genotoxicity of the organic contaminant, since HNPs do not restore control conditions in B(a)P-exposed tissues. 

Regarding Cytome assay results, the potential chromosome-damaging effect of HNP and its ability to face B(a)P genotoxicity have also been explored in vitro. For the first time, an increase of micronucleated cells was assessed in gill cells exposed to HNP alone. In the literature, the ability of carbon-based nanomaterials (carbon nanotubes) to interact with the cellular cytoskeleton, thus inducing errors during the cell cycle in anaphase, has been reported in human cell lines [60]. In agreement with the present results, CB has also been found to be able to induce chromosomal damage in murine macrophage cell lines [61].

Data from the literature support MN and NA evaluation as one possible tool in defining the classification of CB carcinogenicity. As a consequence, CB has been stated by the IARC to be possibly carcinogenic to humans due to the inconclusive results obtained in human cell models (Group 2B) [62]. However, data from CB Cytome studies have been controversial, and the majority of studies have been performed with reference to mammalian cell lines. For example, with respect to murine cell lines, conflicting results emerged from two different in vitro studies performed using the same range of doses and times of exposure [63,64]. With respect to bare CB, the few available data, often limited to human cell lines, were in some cases falsified by CB NPs agglomeration on/inside the exposed cells. This fact worsens the possibility of deeply discussing the results obtained in the present study in a comparative fashion, especially given that the agglomeration behaviors of CB and HNP are quite different (the hydrophilic nature of HNP makes them quite insensitive to agglomeration in ASW). It is worthy of note that, at present, no data are available concerning the carcinogenic of hydrophilic CB-derived NPs, so these findings help to fill some gaps in the knowledge of potential HNP cellular insult. 

TiO_2_ NPs have recently been shown to adsorb several PAHs from soil and water [17], and thus the potential of TiO_2_ NPs to protect against the B(a)P-induced toxicity in biological system appears to be of extreme interest.

In our study, two crystalline forms of nano-TiO_2_ were tested, namely, pure anatase (100%) as MT (as determined by XRD technique and reported in the Results) and Aeroxide^®^ TiO_2_ P25, composed of 70% anatase and 30% rutile, as P25. The two materials differ not only with respect to polymorphic phase composition, but also in terms of SSA values, since the MT sample has a higher SSA than the P25 (60 m^2^ g^−1^). Notwithstanding the different SSA values, both P25 and MT have been shown to quickly agglomerate in both MilliQ W and, more markedly, in ASW, in this case reaching a micrometric size, which was in agreement with previous studies [65]. Such peculiar behavior occurring in high ionic strength media such as seawater (35–40‰) is probably driven by ionic species (released by mineral salts), such as, for instance, divalent cations (i.e., Ca, Mg) adsorbing onto nanoparticle surface and able to neutralize their negative surface charges. As a consequence of such strong aggregation, P25, for instance has been shown to quickly settle in a concentration-dependent manner in ASW, with high removal (up to 80%) of particles suspended in few hours [37]. As such, it could be more prone to interacting with the plasma membrane of cells, thereby inducing a cellular response even at the DNA level.

As a first general overview, data from literature related to TiO_2_-NPs highlight that different experimental conditions and target cell types [19] have different biological outcomes (Appendix A), while superficial morphologies are responsible for different levels of reactivity. The importance of surface morphology has been widely demonstrated [66], underscoring that dimension, shape, crystalline form and surface coating can influence the ability of nanoparticles to induce genotoxicity through direct or indirect actions. Moreover, the doses and amounts of metals they contain are fundamental aspects that are able to modulate nano-TiO_2_ toxicity [67]. Thus, differences with respect to genotoxicity and the capacity to reduce B(a)P genotoxic potential between the two nano-TiO_2_ crystalline forms could be explained with reference to their different nominal compositions, in accordance with the findings of Uboldi and co-workers [68], where rutile was found to be slightly more toxic than anatase. All of these considerations support the interpretation of the data presented here, where a difference was highlighted in terms of genotoxic effect between the different powders tested.

In fact, P25 NPs alone, composed of 30% rutile, caused a loss of DNA integrity. Among the few data available in the literature, D’Agata and co-workers [69] showed that concentrations of nano-TiO_2_ on the order of 10 mg/L resulted in moderate gill DNA damage and hemocyte infiltration. Similarly, many studies on fish have revealed that nano-TiO_2_ can induce oxidative stress, cell membrane damage, protein inactivation and chromosome damage [70]. On the contrary, Della Torre and colleagues [37], after 96 h exposure to 0.1 mg/L, did not find any genotoxic effect exerted by nano-TiO_2_ in mussel gill cells, investigated through Comet assay. 

Since the interaction of NMs with cells can be regarded as a first step in the induction of cellular responses, in vitro studies have focused on elucidating the uptake and biological effects of nano-TiO_2_, suggesting an indirect mechanism as being responsible for the reported genotoxicity [71]. In the present study, the results obtained with the Comet assay were not replaced following the Cytome assay analyses. This difference could be due to nano-TiO_2_ mechanisms of action, since nano-TiO_2_ is characterized by high reactivity, and exerts genotoxic potential mostly through free radical induction [72]. The pro-oxidant effect of free radicals is known to give rise to DNA primary damage which, in turn, might have been repaired under the present experimental conditions, not allowing the expression of stable chromosomal damage. It is thus likely that due to the dimension of the NPs used, they were unable to cross the nuclear pores, resulting in the lack of any NPs observed in the nucleus of the exposed cells. As shown by the TEM images, the genotoxic effects observed come from indirect mechanisms [73]. With respect to the B(a)P exposure data, it is possible that the reason a DNA primary damage but not a chromosomal mutagenic damage was detected might be related to direct exposure to B(a)P (without metabolic activation) applied under the present experimental conditions, which did not allow B(a)P to exert its maximum genotoxic potential [74]. This is probably the reason a DNA primary damage was highlighted. Incidentally, when considering the results of total NA, they showed a statistically significant increase in co-exposure samples with respect to the control. This specific mutagenic potential exerted by P25 might be due to the fact that it is 30% composed of rutile, and this might explain the higher induction of NA observed, compared with the MT data. The results for pure anatase as MT draw a different scenario. First of all, it was found that NPs themselves were harmless in terms of both primary and chromosomal damage. Moreover, after co-exposure, the genotoxic potential of B(a)P was lowered to the control levels. The differences in terms of genotoxicity and capacity to reduce B(a)P genotoxic potential between the two nano-TiO_2_ crystalline forms could be explained with reference to their different nominal composition, as discussed previously. Moreover, their different superficial morphology, highlighted by TEM analysis of the suspended free powders, probably played a key role. Regardless of the differences observed in the dimension of agglomerates between MT and P25 in ASW, their chemical composition seems to be the main driver of the observed cytotoxicity and remediation capability. Therefore, pure anatase was not only confirmed to be genotoxicity-free; it was also shown to be capable of erasing B(a)P effects, at least in vitro, and when using the present experimental model. Even if the in vitro results presented cannot directly explain the effect on the whole organism, their predictive role suggests a good performance of MT in limiting the impact of the genotoxic agent B(a)P on the DNA integrity of gill tissues in marine mussel, at least with respect to the doses and experimental conditions selected. Moreover, although a cytoplasmic rarefaction was observable in all exposed samples compared to the controls, MT-exposed gill biopsies did not show the large cytoplasmic vacuoles detected in the HNP-treated samples. These results do not mean that MT is to be preferred as being considered environmentally safe, because our data are limited to one aspect of the potential interaction between MT and the biota and other dose ranges and exposure times need to be explored, as well as a deep energy dispersive X-ray (EDX) analysis to better define the nature of the particle uptaken. Moreover, further studies are needed to see whether MT particles can interact with other classical pollutants like heavy metals. 

## 5. Conclusions

The present in vitro work indicated pure anatase mesoporous titania (MT powder) to be a cyto- and genotoxicity-free nanomaterial that was able to reduce genetic and chromosomal damage associated with environmental B(a)P exposure. On the contrary, P25 resulted in DNA integrity loss and nuclear abnormalities per se, even if it was able to reduce B(a)P genotoxicity; HNP was responsible per se for the induction of micronucleated cells. 

The contemporary use of two different genotoxicity endpoints allowed for a better clarification of the mechanisms underlying nano-genotoxicity. In addition, TEM ultrastructural investigation suggested an indirect mechanism of action exerted by NPs in the experimental model investigated. 

Moreover, the novelty of the present experimental approach lies in the use of gill biopsies from marine mussels as a quick and useful method for developing in vitro laboratory investigations in eco-nanogenotoxicology. They represent a sort of bridge between the cellular and the organism level, and this is the first time that such an approach has been used to provide preliminary information about the use of organic and inorganic NPs to address the genotoxic impact associated with B(a)P exposure.

## Figures and Tables

**Figure 1 nanomaterials-11-01309-f001:**
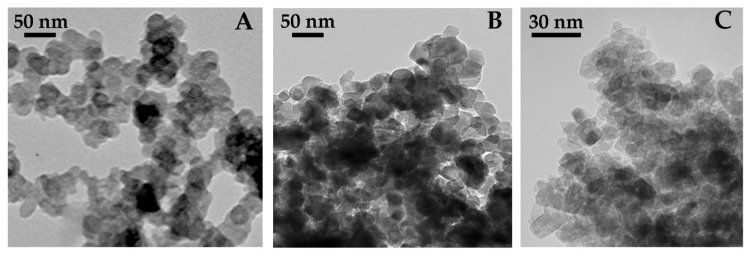
Images obtained by transmission electron microscopy (TEM) of NPs suspended at room temperature (23 °C) through sonication. (**A**) CB-derived hydrophilic NPs (HNP, 50 µg/mL) in distilled water showing aciniform aggregates of almost spherical primary particles; (**B**) Aeroxide^®^ TiO_2_ P25 (P25, 10 µg/mL) in ASW showing large aggregates; (**C**) mesoporous titania (MT, 10 µg/mL) in ASW showing large NPs aggregates.

**Figure 2 nanomaterials-11-01309-f002:**
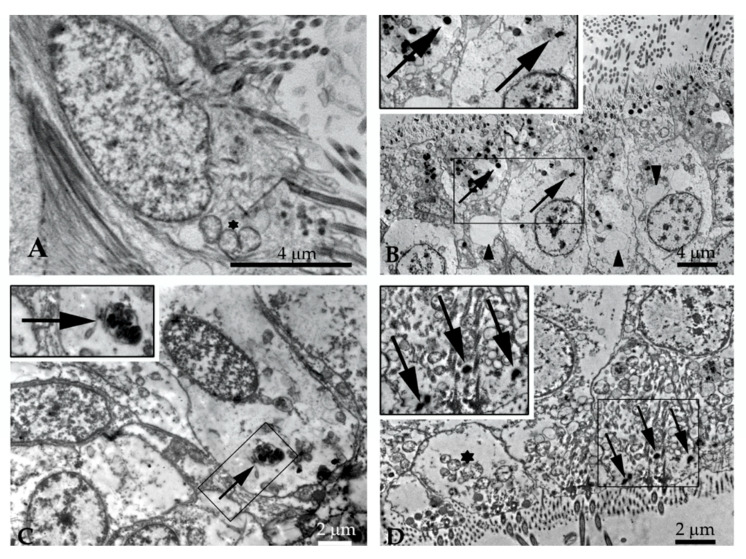
Images obtained by transmission electron microscopy (TEM) of *M. galloprovincialis* gill biopsy exposed to NPs. (**A**) Control. (**B**) 10 μg/mL Hydrophilic CB-derived nanoparticles (HNP). (**C**) 50 μg/mL of P25. (**D**) 50 μg/mL of MT. Magnified fields with arrows indicate electron-dense particles in exposed cells. Arrowhead points to cytoplasmic vacuoles. 
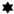
 indicates mitochondria.

**Figure 3 nanomaterials-11-01309-f003:**
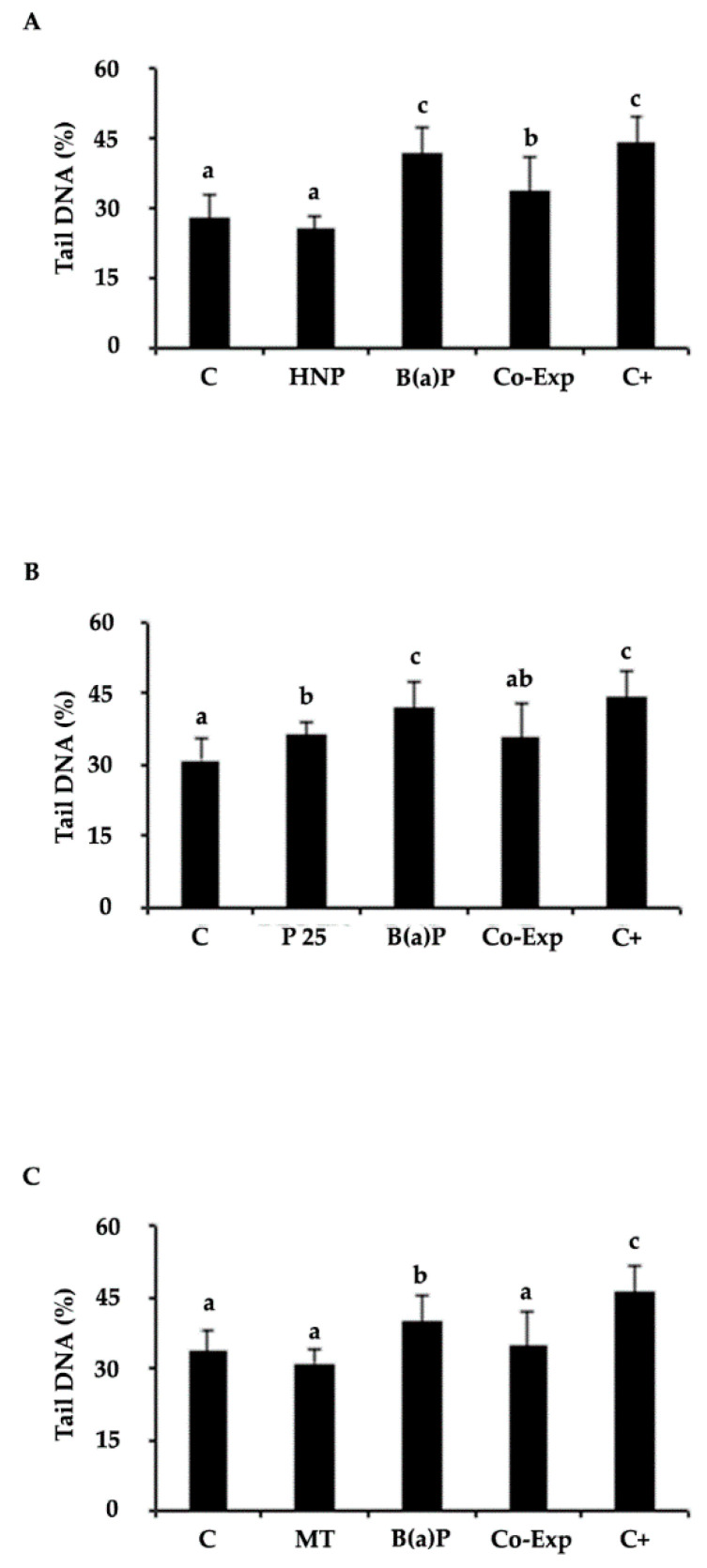
DNA primary damage (% tail DNA) in gill cells of M. galloprovincialis after exposure to the following experimental groups: C = control (ASW); HNP = 10 µg/mL Hydrophilic CB-derived nanoparticles; P25 = 50 µg/mL Aeroxide^®^ TiO_2_ P25; MT = 50 µg/mL Mesoporus titania; B(a)P = 2 μg/mL Benzo(a)Pyrene and NPs, in co-exposure with B(a)P. (**A**) Co-exp: B(a)P and HNP. (**B**) Co-exp: B(a)P and P25. (**C**) Co-exp: B(a)P and MT. C+ positive control (H_2_O_2_ 100 µM). Values are mean ± SD. Different letters indicate significant difference between the groups (Multiple Range Test, MRT *p* < 0.05, n = 9 for each experimental group).

**Figure 4 nanomaterials-11-01309-f004:**
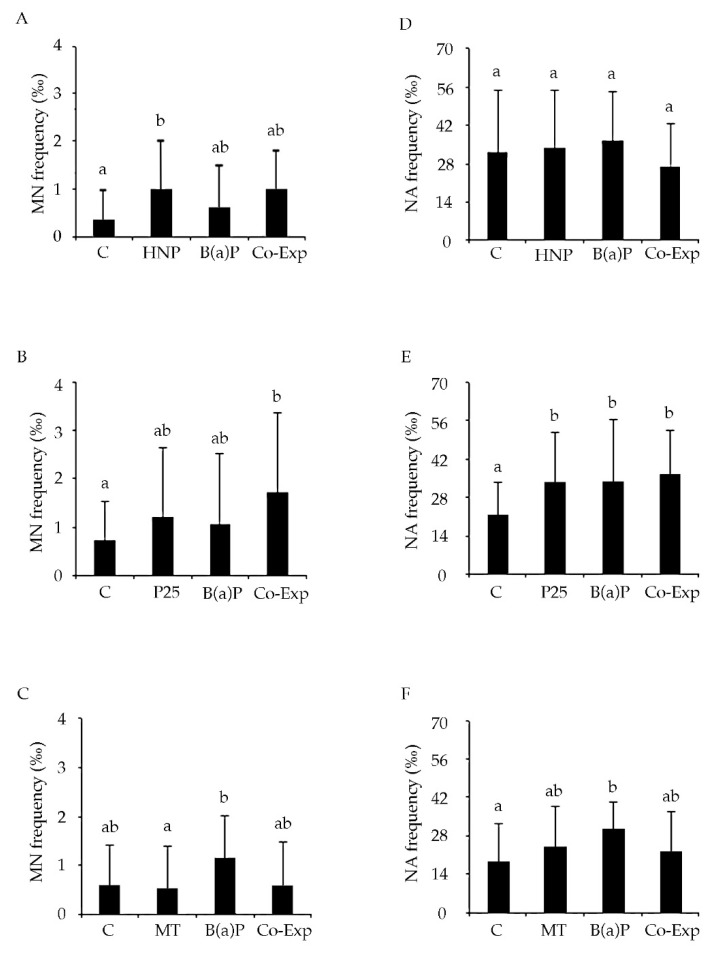
Chromosomal damage (MN and NA frequency) evaluated in gill cells of *M. galloprovincialis* after exposure to the following experimental groups: C = control (ASW); HNP = 10 µg/mL Hydrophilic CB-derived nanoparticles; P25 = 50 µg/mL Aeroxide^®^ TiO_2_ P25; MT = 50 µg/mL Mesoporus; B(a)P = 2 μg/mL Benzo(a)Pyrene; and NPs, in co-exposure with B(a)P. (**A,D**) Co-exp: B(a)P and HNP. (**B,E**) Co-exp: B(a)P and P25. (**C,F**) Co-exp: B(a)P and MT. Values are mean ± SD. Different letters indicate significant difference between the groups (Multiple Range Test, MRT *p* < 0.05, n = 9 for each experimental group).

**Table 1 nanomaterials-11-01309-t001:** Physicochemical characterization by DLS analysis of MT (10 µg mL^−1^) and P25 (10 µg mL^−1^) and HNP in ultrapure water (MilliQ W), ASW at room temperature (23 °C), showing the size-related parameters of NPs, such as Z-Average (nm) and Polydispersity Index (PDI, dimensionless).

	Medium	Z-Average (nm)	PDI
HNP	MilliQ W	165 ± 10	0.12
	ASW	190 ± 10	0.15
P25	MilliQ W	163 ± 9	>0.300
	ASW	972 ± 35	>0.300
MT	MilliQ W	343.4 ± 22.6	>0.300
	ASW	4190 ± 1525	>0.500

## Data Availability

Data is contained within the article.

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
