# Peer review of "Suitability of Nanoparticles to Face Benzo(a)pyrene-Induced Genetic and Chromosomal Damage in M. galloprovincialis. An In Vitro Approach"

_nanomaterials, 2021, doi:10.3390/nano11051309_

Round 1

Reviewer 1 Report

Bernardeschi et al., studied to reduce genetic and chromosomal damages caused by nanoparticles in M. galloprovincialis. The authors performed alkaline comet assay and micronuclei formation to monitor the genotoxic effects. However, the overall quality of the study is not enough for publication.

In addition, I am also not sure in the study concept as to how nanoparticles cause DNA damages to further cause DNA strand breaks and micronuclei formation. Sounds like very indirect effect. The authors must include more direct effects.

Specific comments

1: Although the authors claimed genotoxic effects of the nanoparticles, the only read out is alkaline comet assay and micronuclei formation. Those are not enough. Even in those they must show the representative images with statistical analyses. Without those, we cannot accept the data. In addition, the authors should at least show gamma-H2AX signals with either Western blot or immunofluorescent microscope.

2: At least in mammal, we could imagine that nanoparticles primarily induce inflammation responses, under which DNA damages are accumulated in the resulting cells. A question is how DNA damages are induced in M. galloprovincialis.

3: In spite of the purpose that is to reduce the toxicity of nanoparticles, the study is not performed enough to evaluate the effects, as N numbers and statistical analyses were not clearly indicated. Even in the statistical analyses, it is not clearly shown which condition is better and how much that was different.

Author Response

While thanking the Reviewer for the suggestions, here below The Authors report a list of the changes made to improve the manuscript:

Bernardeschi et al., studied to reduce genetic and chromosomal damages caused by nanoparticles in M. galloprovincialis. The authors performed alkaline comet assay and micronuclei formation to monitor the genotoxic effects. However, the overall quality of the study is not enough for publication.

 In addition, I am also not sure in the study concept as to how nanoparticles cause DNA damages to further cause DNA strand breaks and micronuclei formation. Sounds like very indirect effect. The authors must include more direct effects. The Authors agree with the Reviewer about the importance to study the direct effect of NPs on cellular components, anyway it was not the aim of the present work which was actually to investigate the harmlessness of the selected NPs in terms of DNA primary damage and chromosomal damage in marine mussel biopsies, as well as NPs capability in recovering organic toxic insult after in vitro B(a)P exposure, as reported in the introduction section. On the other hand, the importance of indirect effects of NPs in inducing genotoxicity is widely recognized (Magdalevova et al., 2014)

Specific comments

1: Although the authors claimed genotoxic effects of the nanoparticles, the only read out is alkaline comet assay and micronuclei formation. Those are not enough. Even in those they must show the representative images with statistical analyses. Without those, we cannot accept the data. If the Referee means the representative images of Comet nuclei and micronuclei a new figure on MN has now been added in the Supplementary Materials (see paragraph 2.5.Figure S2). Moreover, the figure legend referring to SM Comet image has been expanded. In addition, the authors should at least show gamma-H2AX signals with either Western blot or immunofluorescent microscope. The presence of double strand breaks has been actually included in the present work through the alkaline version of Comet assay. In fact, during the electrophoresis run carried out at pH 13 both single and double strand breaks are detected, besides alkali labile sites (Tice et al., 2000) as now specified on paragraph 2.4.

2: At least in mammal, we could imagine that nanoparticles primarily induce inflammation responses, under which DNA damages are accumulated in the resulting cells. A question is how DNA damages are induced in M. galloprovincialis. The Authors agree and thank the Reviewer for the comment. As the nanoparticles used do not possess dimensions allowing them to enter the nucleus the Authors believe that the effects observed are of indirect origin, as now better specified in the text in the discussion section.

3: In spite of the purpose that is to reduce the toxicity of nanoparticles, the study is not performed enough to evaluate the effects, as N numbers and statistical analyses were not clearly indicated. N numbers have been added in the figure legends. Even in the statistical analyses, it is not clearly shown which condition is better and how much that was different. A sentence on the different performances of nanomaterials in terms of statistical strength has been provided in the 3.3.1. paragraph, where it is now stated that MT powder is the only one to statistically restore B(a)P exposed cells to control levels.

Reviewer 2 Report

This work investigates the genetic and chromosomal damage effects of benzo(a)pyrene and various nanoparticles (which could be used for its remediation) in M.galloprovincialis.

The topic is important and fits well within the scope of the journal. The paper is well-written, the results are sound and supported by the presented data. I can support the publication of this paper in Nanomaterials after addressing some mostly minor issues:

1) There are some mistakes in the use of English, please carefully revise the manuscript (e.g., page 3 line 120, “chemicals volumes” -> volume of the chemicals, or chemicals’ volume; similar in page 4 line 191 “Particles behavior”). Also: “Statistical analysis was performed by the use of software SGWIN(Windows98) was used.”

2) Please use decimal points and not commas (, ->.). (e.g. “p<0,05”, or in Fig. 4).

3) There are a couple of small issues with Fig. 4. In A) and D) CB label is used instead of HNP (which is in the caption and in Fig. 3). Also, the Y-axis indicates NA frequency while the caption says AN frequency. Letters indicating the significance groups are missing from D). Also, I suggest modifying the Y-range of A) and C) from 0 to 4, to match B). There might also be a mistake in the significance labels in C). Bars C and MT are nearly identical, yet they have different labels (a and ab), while the significantly higher B(a)P has also ab. I suggest double-checking it.

4) Please try to be less general in the conclusions and more precise. The last sentence of the conclusions is repeated from the abstract “The present in vitro approach based on the use of marine mussel…”. Please be specific about these commonly confused and controversial aspects.

5) I also suggest reconsidering/rephrasing the first sentence of the Conclusions. Two of three nanoparticles were proven to be genotoxic based on the presented data, yet the authors suggest more tests to “support their safe application to reduce genetic and chromosomal damage associated with environmental B(a)P exposure”. I understand the intention of the authors, but I find the phrasing a bit controversial.

Author Response

While thanking the Reviewer for the suggestions, here below The Authors report a list of the changes made to improve the manuscript:

This work investigates the genetic and chromosomal damage effects of benzo(a)pyrene and various nanoparticles (which could be used for its remediation) in M.galloprovincialis.

The topic is important and fits well within the scope of the journal. The paper is well-written, the results are sound and supported by the presented data. I can support the publication of this paper in Nanomaterials after addressing some mostly minor issues:

1) There are some mistakes in the use of English, please carefully revise the manuscript (e.g., page 3 line 120, “chemicals volumes” -> volume of the chemicals, or chemicals’ volume; similar in page 4 line 191 “Particles behavior”). Also: “Statistical analysis was performed by the use of software SGWIN(Windows98) was used.” The Authors thank the Reviewer for his/her comments, assuring that the mistakes indicated have been corrected accordingly;

2) Please use decimal points and not commas (, ->.). (e.g. “p<0,05”, or in Fig. 4). Decimal points are now used in Fig. 4;

3) There are a couple of small issues with Fig. 4. In A) and D) CB label is used instead of HNP (which is in the caption and in Fig. 3). Also, the Y-axis indicates NA frequency while the caption says AN frequency. Letters indicating the significance groups are missing from D). Also, I suggest modifying the Y-range of A) and C) from 0 to 4, to match B). There might also be a mistake in the significance labels in C). Bars C and MT are nearly identical, yet they have different labels (a and ab), while the significantly higher B(a)P has also ab. I suggest double-checking it. The Authors thank the Reviewer; HNP is now used in Fig. 4, AN frequency has been corrected in the caption. Fig. 4 D now shows letters indicating the significance groups. Y-range of A) and C) is now from 0 to 4 as requested. The Authors thank the Reviewer for the suggestion and the correct statistical significant levels are now reported.

4) Please try to be less general in the conclusions and more precise. The last sentence of the conclusions is repeated from the abstract “The present in vitro approach based on the use of marine mussel…”. Please be specific about these commonly confused and controversial aspects. Conclusion section has been modified according to the Referee’s suggestions

5) I also suggest reconsidering/rephrasing the first sentence of the Conclusions. Two of three nanoparticles were proven to be genotoxic based on the presented data, yet the authors suggest more tests to “support their safe application to reduce genetic and chromosomal damage associated with environmental B(a)P exposure”. I understand the intention of the authors, but I find the phrasing a bit controversial. The first sentence of the Conclusions has been also modified accordingly.

Reviewer 3 Report

This study is interesting, quite well structured, but my biggest concern is with the originality and novelty of the study. 

  • The novelty and originality of the study is not clear. From the information presented in the manuscript it is observed that all the nanomaterials used were synthesized and characterized according to protocols from the literature. In these conditions, the authors must bring more clarifications in this regard. The degree of originality and novelty must be explicitly mentioned. In addition, there are too many bibliographic references for an original article. The authors should review this aspect, especially in the experimental part and in the results and discussions section.
  • Other observations:
    • The information in the introduction is not linked but only presented in a telegraphic manner. The whole section must be reviewed and the correlations must be made between the subsections.
    • The images in Figure 1 should be presented side by side so that they can be better compared by readers.
    • The quality of the images included in Figure 4 needs to be improved. The text is too small and difficult to read.
    • The discussion section compares the results obtained by the authors with other studies in the literature. a comparative table would be beneficial for readers here.

Author Response

While thanking the Reviewer for the suggestions, here below The Authors report a list of the changes made to improve the manuscript:

REVIEWER 3

This study is interesting, quite well structured, but my biggest concern is with the originality and novelty of the study.

The novelty and originality of the study is not clear. From the information presented in the manuscript it is observed that all the nanomaterials used were synthesized and characterized according to protocols from the literature. In these conditions, the authors must bring more clarifications in this regard. The degree of originality and novelty must be explicitly mentioned. In addition, there are too many bibliographic references for an original article. The authors should review this aspect, especially in the experimental part and in the results and discussions section. The Authors thank the Referee for his/her suggestion: now the novelty and originality of the study have been underlined in the introduction and discussion sections. To our knowledge, this is the first time that the safety and the efficacy of specifically designed nanomaterials in preventing genetic and chromosomal damage induced by B(a)P was investigated through an in vitro nano-remediation approach. Moreover, the Authors clarified that many of the numerous references reported in the experimental part are not just taken from the literature but they belong to the working group who first designed and synthetized nanomaterials with specific characteristic to be used in the nano-remediation field, then published chemical characterization data, applied them to in vitro experimental model and, at the very end, wrote results obtained in cells.

Other observations:

The information in the introduction is not linked but only presented in a telegraphic manner. The whole section must be reviewed and the correlations must be made between the subsections. The introduction section has been reviewed according to the Referee’s suggestion.

The images in Figure 1 should be presented side by side so that they can be better compared by readers. Figure 1 has been changed according to the Referee’s suggestion.

The quality of the images included in Figure 4 needs to be improved. The text is too small and difficult to read. The Authors thank the Referee for his/her suggestion: the text in Fig. 4 has been enlarged in order to allow a better reading.

The discussion section compares the results obtained by the authors with other studies in the literature. a comparative table would be beneficial for readers here. A comparative table has been added as requested by the Reviewer in the Supplementary Materials and cited in the discussion section .

Round 2

Reviewer 1 Report

This is a revised prepared by Bernardeschi et al. I am appreciated for the revision. However, the conceptual points of this manuscript are still not clear. I feel the author did not really revised the manuscript.

Specific comments

1: In the title of the manuscript, this study is like focusing on the reduction of genetic and chromosomal damages. However, this manuscript is rather simply studying on the effects of the nanoparticle exposure to M. galloprovincialis.

What exactly is the point to see the effects of the nanoparticle exposure of environmental nanoparticles?

What exactly is “to reduce genotic and chromosomal damage” on the title meaning? If what the authors really meant is “reduction of genotic and chromosomal damages by M. galloprovincialis”. Then, the authors must design the study to see the environmental toxicity reduction. If not, the authors must clarify the point.

2: Although the authors claimed genotoxic effects of the nanoparticles, the only read outs are alkaline comet and micronuclei formation assays, in which the differences are not very clear. The authors must add other experiments that enable to see the statistical significance. My recommendation is to see gamma-H2AX signal by western and/or immune-fluorescent microscope.

Author Response

Response to Reviewer 1

1: In the title of the manuscript, this study is like focusing on the reduction of genetic and chromosomal damages. However, this manuscript is rather simply studying on the effects of the nanoparticle exposure to M. galloprovincialis. The Authors apologize if they made their research focus not completely clear. They actually aimed at finding not-genotoxic NPs to be used in co-exposure experiments with the known genotoxic compound B(a)P to see if such NPs were able to reduce the genotoxic effects caused by B(a)P in marine mussel gill biopsies. A more detailed explanation has now been added in the introduction section and the title was changed, according to the Reviewer suggestion, to make it clearer.

What exactly is the point to see the effects of the nanoparticle exposure of environmental nanoparticles? The NPs used have been selected only to be used in co-exposure experiments. For that reason they should not be genotoxic and their potential genotoxicity was thus assessed. Only one of the three NPs used was found not to be genotoxic and able to reduce genotoxic effects induced by B(a)P.

What exactly is “to reduce genotic and chromosomal damage” on the title meaning? If what the authors really meant is “reduction of genotic and chromosomal damages by M. galloprovincialis”. Then, the authors must design the study to see the environmental toxicity reduction. If not, the authors must clarify the point. The Authors thank the Reviewer for underlining the need to clarify the title which did not mention B(a)P. We believe that the new title “Suitability of nanoparticles to face B(a)P-induced genetic and chromosomal damage in M. galloprovincialis. An in vitro approach” makes the study design clearer.

2: Although the authors claimed genotoxic effects of the nanoparticles, the only read outs are alkaline comet and micronuclei formation assays, in which the differences are not very clear. The authors must add other experiments that enable to see the statistical significance. My recommendation is to see gamma-H2AX signal by western and/or immune-fluorescent microscope. The Authors apologize but they are not sure to understand the question.  Well, in case the Reviewer means that the two endpoints did not reveal strong genotoxic effects after NPs exposure, the aim of the study was to actually find not-genotoxic NPs. If he/she meant that the two selected endpoints are not the right ones to study B(a)P genotoxicity a new part has been added in the introduction section. It is reported that the Comet and the Micronucleus assays are genotoxicity biomarkers widely used in aquatic organisms. It is explained that the Comet Assay is used to microscopically detect DNA damage asDNA single strand breaks, double strand breaks and alkali labile sites at single cell level. Micronucleus assay, on the other hand, provides an efficient measure of chromosomal DNA damage (Bolognesi e Fenech 2012). Thus, the Comet assay detects primary repairable, therefore reversible DNA lesions (alkali labile sites, single strand DNA breakages), the MN assay detects more persistent DNA lesions or aneugenic effects, result of either chromosome breakage or chromosome mis-segregation during mitosis, that cannot be repaired (Bombail et al., 2001, Hartmann et al., 2001). Concerning a deeper study on double strand breaks by Western and/or immune-fluorescent microscope it might be surely interesting but it wasn’t our scope at the moment and we couldn’t do that now because of the problems we have had with our laboratory facilities after Covid-19 emergency.

Reviewer 3 Report

I carefully read the authors' answer to my observations and the revised version of the manuscript and I have noticed a significant improvement in the quality of the study. However, I am still not convinced by the authors' answer regarding the novelty of the study or the justification of the big number of references from the experimental part and the discussion part of the manuscript. I believe that this should be revised by the authors.

Author Response

Response to Reviewer 3

I carefully read the authors' answer to my observations and the revised version of the manuscript and I have noticed a significant improvement in the quality of the study. However, I am still not convinced by the authors' answer regarding the novelty of the study or the justification of the big number of references from the experimental part and the discussion part of the manuscript. I believe that this should be revised by the authors. We do thank the Reviewer for his/her suggestions. The novelty of the study has been now re-written both in the introduction and conclusions sections. Moreover, six references have been removed from Materials and Methods section and other four ones  from the discussion section. The table in Supplementay Material has been modified accordingly.

Round 3

Reviewer 1 Report

This is a 2nd revised manuscript. I am appreciated for the revision. I now have somehow better understand on the authors’ study. However, I would recommend to re-write the comprehensive abstract with clearly stating the question/problem what the authors are really addressing, after the brief background explanation with a few sentences. Explanations of the major findings can be in the following. Otherwise, it is hard for the reader to get what the authors are really addressing with a series of experiments.

Regarding chromosomal damage detection, I still have to argue the potential problem in the methods what the authors used. In fact, although alkaline comet assay is a conventional method, those give quite high background in some cases. Therefore, this technique is not very confident by itself: I agree that it gives some circumstantial evidence, though. In addition, although the authors also monitored micronuclei, those usually show transient arise when it is damaged. If the authors could do those experiments with time cause, that will be better.

Although the authors cannot perform western and immunofluorescent microscope analyses in their current the authors’ situation. I am just telling my recommendation. If the authors can be enough confident on those methods, paticularly in this study. I am fine with that.

Author Response

This is a 2nd revised manuscript. I am appreciated for the revision. I now have somehow better understand on the authors’ study. However, I would recommend to re-write the comprehensive abstract with clearly stating the question/problem what the authors are really addressing, after the brief background explanation with a few sentences. Explanations of the major findings can be in the following. Otherwise, it is hard for the reader to get what the authors are really addressing with a series of experiments. The Authors thank the Reviewer for his/her suggestions. The abstract was re-written following the Reviewer’s reccomandation. In particular, the aim of the study is now better explained.

Regarding chromosomal damage detection, I still have to argue the potential problem in the methods what the authors used. In fact, although alkaline comet assay is a conventional method, those give quite high background in some cases. Therefore, this technique is not very confident by itself: I agree that it gives some circumstantial evidence, though. In addition, although the authors also monitored micronuclei, those usually show transient arise when it is damaged. If the authors could do those experiments with time cause, that will be better. In order to clarify the meaning of the biomarkers used, the Authors modified the subsections 2.4 and 2.5 titles and the abstract. Indeed, the Comet assay detects primary, repairable and therefore reversible DNA lesions, while Cytome assay gives an efficent measure of chromsomal damage. It detects persistent, fixed damage, which is not repairable. On the other hand, the Authors agree with the Reviewer on the usefulness of longer exposure times which actually belong to our future perspectives, as well as an in-depth analysis of double strand breaks induction. Regarding the high background levels detected with alkaline Comet assay, the Authors thank the Reviewer for his/her observation. Such an aspect had been reported in the Discussion section, even underlining how the basal DNA damage level observed in control cells still guarantees the detection of statistically significant differences if compared with exposed ones.

Although the authors cannot perform western and immunofluorescent microscope analyses in their current the authors’ situation. I am just telling my recommendation. If the authors can be enough confident on those methods, paticularly in this study. I am fine with that. The Authors are sorry for not being able to preform the suggested experiments at the moment but it will be of great interest in the next future.